# LanguageRefer: Spatial-Language Model for 3D Visual Grounding

**Junha Roh, Karthik Desingh, Ali Farhadi, Dieter Fox**
Paul G. Allen School, University of Washington, United States
{rohjunha, kdesingh, ali, fox}@cs.washington.edu

**Abstract:** For robots to understand human instructions and perform meaningful tasks in the near future, it is important to develop learned models that comprehend referential language to identify common objects in real-world 3D scenes. In this paper, we introduce a spatial-language model for a 3D visual grounding problem. Specifically, given a reconstructed 3D scene in the form of point clouds with 3D bounding boxes of potential object candidates, and a language utterance referring to a target object in the scene, our model successfully identifies the target object from a set of potential candidates. Specifically, LanguageRefer uses a transformer-based architecture that combines spatial embedding from bounding boxes with fine-tuned language embeddings from DistilBert [1] to predict the target object. We show that it performs competitively on visio-linguistic datasets proposed by ReferIt3D [2]. Further, we analyze its spatial reasoning task performance decoupled from perception noise, the accuracy of view-dependent utterances, and viewpoint annotations for potential robotics applications. Project website: https://sites.google.com/view/language-refer.

**Keywords:** Referring task, Language model, 3D visual grounding, 3D Navigation

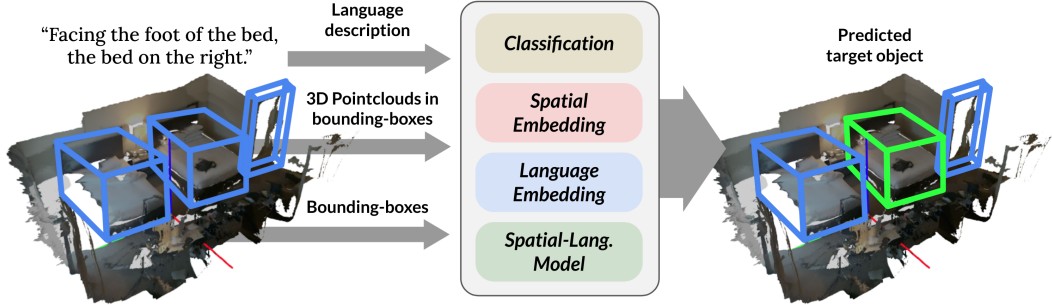

Figure 1: **Simplified overview of LanguageRefer.** The **LanguageRefer** model takes as input a grounding language description of a single object in the scene, a 3D point cloud of a scene, and bounding boxes of objects in the scene and predicts the target object. Its four modules include: a classifier, a spatial embedder, a language embedder, and a spatial-language model.

## 1 Introduction

For robots to communicate seamlessly with humans to perform meaningful tasks in indoor environments, they must understand natural language utterances and ground them to real-world elements. Several recent advances have combined language and visual elements, producing methods for tasks such as visual question and answering (VQA) involving spatio-temporal reasoning tasks [3, 4, 5], embodied QA [6], and pre-training for visual recognition tasks with language descriptions [7]. Further, embodied agents can follow visually grounded language instructions to perform embodied tasks [8, 9]. However, for real robots to intelligently perform these tasks, we need 3D representations from raw sensor data; ReferIt3D [2] proposes a benchmarking dataset of language utterances referring to objects in 3D scenes from the ScanNet [10] dataset.

5th Conference on Robot Learning (CoRL 2021), London, UK.

Our long-term goal is to enable robots to visually navigate indoor environments based on referential instructions. In this paper we take a step towards this goal by leveraging the ReferIt3D dataset to build a model that can identify the 3D object referred to in a language utterance.

Referential language to identify an object in real-world 3D scenes poses a challenging problem. Consider the sample utterance *"Facing the foot of the bed, the bed on the right"* along with a 3D scene as shown in Figure 1. Humans can easily follow the language clues, infer the point of view of the speaker, locate all the referenced elements, and spatially reason to locate the *bed* in the scene despite two instances of *beds*. However, viewpoint prediction, object identification, and spatial reasoning remain open-ended research problems in robotics and vision. The 3D reference task proposed by ReferIt3D is difficult because: (1) reconstructed 3D scenes of the real world in the ScanNet dataset are noisy and lack fine details compared to 2D images or rendered 3D scenes, (2) fine-grained class labels and expressions in natural language utterances are diverse and not exactly matched, (3) view-dependent utterances often require guessing the original viewpoints, which deters the model from properly learning spatial concepts, and (4) the combined complexity of multiple challenges complicates efforts to analyze what the model learns or understands.

Inspired by the success of the methods in [11, 12] on CLEVR and CLEVRER domains for the spatial reasoning task, we hypothesized that for the 3D reference task, decoupling the spatial-reasoning from the perceptual task of identifying the objects in the 3D scene would improve performance and clearly track the role of perception noise in performance. More specifically, instead of developing an integrated multi-modal perception system, we assumed a pre-trained instance classification model or ground-truth classes for the objects that informed the spatial reasoning task. We focused on how the language model with spatial information could handle the reference task.

The ReferIt3D dataset includes two sub-datasets containing natural and synthetic language utterances, namely Nr3D and Sr3D, respectively. In our experiments, our model achieved comparable scores on both with predicted instance class labels. We observed high accuracy with ground-truth labels in Sr3D, which indicates that our model better understood template-based language data. Since our pipeline is modular and features multiple models (perceptual, spatial embedding, pre-trained language embedding, and spatial-language), our approach is flexible and adaptable to different environments and object entities.

Another aspect of the 3D reference task is viewpoint prediction. The ReferIt3D dataset contains utterances that can be grouped into view-independent (VI) and view-dependent (VD) categories. An example of a VI utterance is *"The lamp closer to the white armchair"* and a VD utterance is *"The lamp on the right in-between the beds"*. The VD utterance requires viewpoint prediction. This distinction is crucial for robotics applications where the agent must infer the viewpoint to which the speaker is referring. In the ReferIt3D dataset, some VD utterances lack information to guess the valid orientation of the agent (who utters the language description), which prevents a model from understanding the significance of spatial relationships such as 'left of.' This is due to the annotation process of ReferIt3D datasets, where both a speaker and a listener can freely rotate the scene to infer an ill-defined orientation in the utterance. Human annotators who are aware of spatial concepts tend to validate otherwise arbitrary orientations. However, a data-driven model will suffer from degraded learning when it cannot verify orientations as well as human annotators. Viewpoint-free annotations may suit most robotics applications; nonetheless, predicting or verifying whether a given statement and the viewpoint match remains important. To better train and investigate the agent in view dependencies, we provide an extra collection of orientation annotations for VD utterances and compare the models with and without viewpoint correction from them.

To summarize, the paper contributes: (1) a novel transformer-based spatial-language model in a modular pipeline that better understands spatial relationships in a 3D visual grounding task. We show that our model achieves comparable performance with state-of-the-art methods on the Nr3D and Sr3D datasets. (2) analysis of the ReferIt3D dataset with viewpoint orientation annotations to remove potential artifacts from implicit orientations. (3) ablation and additional experiments with ground-truth classes that decouple the impact of perception noise in the spatial reasoning task.

## 2 Related Work

### 2.1 Vision-and-Language Navigation and Robot Navigation

Vision-and-language navigation (VLN) has been extensively studied and made remarkable progress over the last few years ([13, 14, 15, 16, 17, 18, 19, 20, 21, 22, 23]). Given a language instruction in the simulation environment, the goal is for an agent to reach the desired node on the pre-defined traversal graph using images as input. The literature offers several extensions. ALFRED [8] proposes an extended VLN task that grounds a sequence of sub-tasks to achieve a higher level task in the AI2Thor environment [9]. ALFRED navigation sequences have (implicit) goals that are often close to objects of interest in the subsequent sub-tasks, e.g., when an agent is asked to move in front of the sink because it is going to clean a cup there. With high-level semantic tasks, object-centric spatial understanding is of even greater importance.

In another direction, recent approaches have relaxed the constraint of discrete traversal in VLN into continuous space ([24, 25, 26, 27, 28]). Here, an agent encounters more complex tasks involving time and space. Thus, expanding the space representation to 3D can be an effective solution. In the context of VLN, we consider the 3D visual grounding task as a proxy for 3D indoor navigation that includes the full observability assumption and goal-oriented language descriptions. In particular, our approach focuses on understanding spatial relationships among objects, which plays a key role in VLN and robot navigation.

### 2.2 2D and 3D Visual Grounding

The 2D visual grounding task localizes an object or region in an image given a language description about the object or region ([29, 30, 31]). Most methods use two-stage approaches: they first generate proposals and then compare the proposals to the language description to choose the grounded proposal ([32, 33, 34, 35, 36]).

The 3D visual grounding task localizes a 3D bounding box from the point cloud of a scene given a language description. Recently, ReferIt3D [2] and ScanRefer [37] were proposed as datasets for 3D visual grounding tasks, with language annotation on the ScanNet [10] dataset. Most 3D grounding approaches ([2, 37, 38, 39, 40]) follow a two-stage schemes similar to many 2D visual grounding tasks. First, multiple bounding boxes are proposed or the ground-truth bounding boxes are used, and then features from the proposals are combined or compared with features from the language description. InstanceRefer [38] extracts attribute features both from point clouds and utterances and compares them to select the object that best matches. FFL-3DOG [39] matches features from language and point clouds with guidance from language and a visual scene graph. These methods rely on specific designs, e.g., bird-eye-view mappings with different types of operations or intensive language pre-processing for graph generation. *In contrast, our approach leverages the language embedding space from the pre-trained language model.* Following the pipeline and general architecture of the language model therefore requires minimal manual design compared to previous works. SAT [40], like our approach, relies on transformer [41] models; it learns a fused embedding of multi-modal inputs and uses auxiliary 2D images. In contrast, our approach uses a semantic classifier to predict object class labels and takes these labels as input. It has a marginal cost of learning fused embeddings compared to training multiple BERT models [42] from scratch in SAT [40]. Even given only semantic information from point clouds, our model still achieved comparable performance with state-of-the-art methods on Nr3D and Sr3D datasets. In addition, the decoupled perception module makes our approach modular and thus transferable to different data.

## 3 Problem Statement and Methodology

Given a 3D scene $S$ with a list of objects $(O_1, \cdots, O_M)$ and a language utterance $U$, the problem is to predict the target object $O_{\mathcal{T}}, \mathcal{T} \in \mathcal{I}_M = \{1, \cdots, M\}$ referred to in the language. A single object $O_i$ consists of a bounding box $B_i \in \mathcal{B} = \mathbb{R}^6$ and corresponding point cloud $P_i \in \mathcal{P} = \mathbb{R}^{N_i \times 6}$ (xyz positions and RGB values) in the bounding box with $N_i$ number of points.

We propose an approach based on language models, called **LanguageRefer**, to solve a 3D visual grounding task. Our model focuses on understanding spatial relationships between objects from language descriptions and 3D bounding box information. We chose this approach due to (1) the high

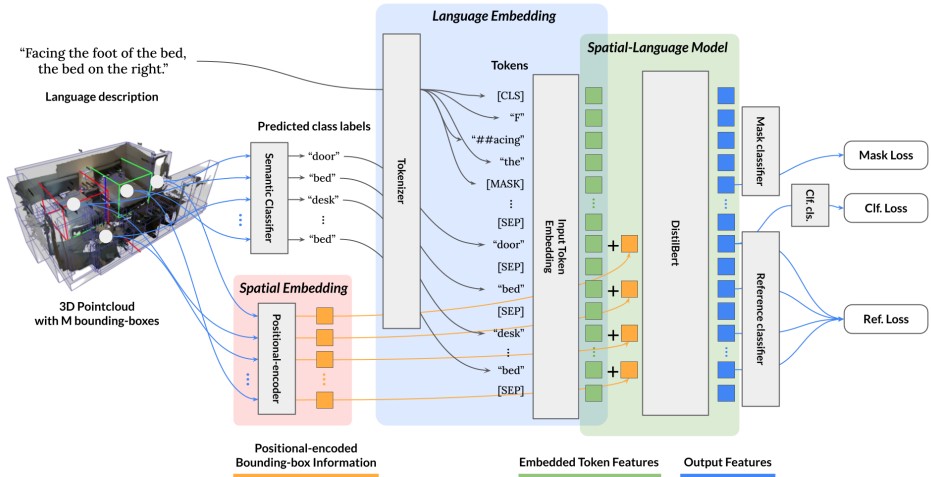

Figure 2: **Detailed overview of LanguageRefer.** A semantic classifier predicts class labels from a 3D point cloud in each bounding box (using color and xyz positions). The language description or utterance (e.g., "*Facing the foot of the bed, the bed on the right*") is transformed into a sequence of tokens. The input token embedding in DistilBert [1] converts the tokens into embedded feature vectors (green squares). Bounding box position and size information are positional-encoded to form encoded vectors using techniques from [41] (orange squares); they are added to the corresponding embedded feature vectors (green squares). After the addition, our reference model processes the modified features and feeds them to multiple tasks. The main task is a reference task, i.e., it chooses the referred object from the object features. The instance classification task is a binary classification, i.e., it determines whether the given object feature belongs to the target class. Finally, the masking task, commonly used in language modeling, recovers the original token from a randomly replaced token in the utterance.

dependency on spatial relationship descriptions in language, and (2) the holistic nature of spatial relationship information, which differs from unary attribute information such as color and shape. As shown in Figure. 2, we use a two-stage approach. First, we determine the class labels of objects in the scene. Second, we use spatial-language embedding to identify the referenced object. The following subsections describe these steps in detail.

**Semantic Classification Model and Tokenization.** For semantic classification of the point cloud in a bounding box, we employed PointNet++ [43], which achieved 69% accuracy on average in the test dataset. In training and inference, we use a sampled point cloud $P'_i \in \mathbb{R}^{1024 \times 6}$ and PointNet++ predicts the semantic label $\hat{l} \in \mathcal{L}$, where $\mathcal{L}$ is a set of class labels in text. Each scene has pairs of predicted class labels and bounding box values $((\hat{l}_i, B_i) : \hat{l}_i \in \mathcal{L}, B_i \in \mathcal{B})$ from objects. Predicted labels are concatenated to the utterance $U$ with a separator [SEP] and then split by a tokenizer into a list of indices of tokens: $U$ becomes $(u_1, \cdots, u_t)$, and each predicted class label $\hat{l}_i$ becomes $(o_1^i, \cdots, o_{n_1}^i)$, where each token index is in $\mathcal{I}_D$ and $D$ is the size of the dictionary.

**Language Model and Token Embedding Generation.** Our model uses a pre-trained language model, DistilBert, for the reference task. Transformers consider relationships among all pairs of elements through attention, and they can be effectively leveraged to explain spatial relationships between objects as discrete entities. In our formulation, predicted class labels are considered to be sentences, so they are concatenated to the utterance with separation by [SEP]. Therefore, the final sequence of token indices would be V = ([CLS], $u_1, \cdots, u_t$, [SEP], $o_1^1, \cdots, o_{n_1}^1$, [SEP], $\cdots$, [SEP], $o_1^M, \cdots, o_{n_M}^M$, [SEP]). Though the token index sequence complies with the specification of DistilBert, it violates the number of sentences. Then, we transform $V$ into the token embedding sequence $W = (w_1, \cdots, w_T), w_i \in \mathbb{R}^{768}$ using DistilBert's word embeddings. For concise notation, we define the indices of utterance tokens in $V$ or $W$ as a mask $M_U = (2, \cdots, t+1)$ and the indices of first tokens from objects $(o_1^1, \cdots, o_1^M)$ as a mask $M_O$. We also define a mask operator $[\cdot]$ to manipulate specific elements in the sequence; for instance, $W[M_U] = 0$ empties all utterance embeddings in $W$.

**Spatial-Language Model and Spatial Embeddings.** To combine spatial information from raw bounding box values into the token embedding $W$, we employ sinusoidal positional encoding $\mathrm{PE}(\cdot)$ from [41] to transform the bounding box vector (center position and size) $B_i \in \mathcal{B} \subset \mathbb{R}^6$ to $b_i =$

$PE(B_i) \in \mathbb{R}^{768}$, which is then added to $W[M_O]$.[1] The token embedding $W$ is then combined with spatial information and finally transformed into the output embedding $X = (x_1, \cdots, x_T), x_i \in \mathbb{R}^{768}$ by the reference model, which is fine-tuned from the pre-trained DistilBert. The final reference task is performed by the reference classifier from $X[M_O]$, as explained in the following subsection.

**Loss Functions**. We use three tasks for training and corresponding loss values. First, we use the reference loss, $\mathcal{L}_{ref}$, following the original proposal in [2]. We ask the model to choose one object as the target instance from $M$ candidates. We collect scalar values from $X[M_O]$ by a linear layer and take the argmax on those values to choose the target instance.

Second, we add a binary target classification loss $\mathcal{L}_{clf}$ on $X[M_O]$ to determine whether a given object belongs to the target class.

Last, we employ mask loss from language model pre-training, $\mathcal{L}_{mask}$. We randomly replace the tokens from nouns in the utterance with a probability of 15 %. The noun token is replaced by [MASK] with an 80 % chance, by a random token with a 10 % chance, or it remains the same with a probability of 10 %. Then, the model is asked to recover the original token index. We expect the model to fill in the replaced tokens in the utterance by understanding the relationship between objects. We use cross entropy loss for all tasks and compute the final loss as

$$\mathcal{L} = \mathcal{L}_{ref} + 0.5\mathcal{L}_{clf} + 0.5\mathcal{L}_{mask}. \tag{1}$$

At inference, we followed the approach of InstanceRefer [38] to filter out objects that do not belong to the predicted target class. We used an extra DistilBert-based target classification model of 94 % accuracy that takes the language utterance as input to predict the target class. To reduce the chance of removing the true target instance in the filtering process, the top-$k$ class predictions (from the semantic classifier) for each object are compared to the predicted target class. We use $k = 4$ throughout the experiments. For masking loss computation, we extracted nouns in the utterance. We used flair [44] for part-of-speech (POS) tagging.

## 4 Experiments

### 4.1 Datasets

We evaluated our model on the reference task from ReferIt3D [2] with two datasets, Nr3D (Natural reference in 3D) and Sr3D (Spatial reference in 3D, which contains spatial descriptions only). Both datasets augment ScanNet [10], a reconstructed 3D indoor scene dataset with language descriptions. Nr3D has 41,503 natural language utterances, and Sr3D contains 83,572 template-based utterances, on 707 scenes following the official splits of ScanNet [10]. The datasets have 76 target classes and are designed to have multiple same-class distractors in the scene.

### 4.2 Experiment Settings

ReferIt3D [2] provides the ground-truth bounding boxes of objects in the scene, point clouds of the scene, utterances and corresponding target objects. We measured the accuracy of the model by comparing the object selected from $M$ candidates to the ground-truth target object. When the number of same-class distractors exceeded two, we classified the instance as "hard" according to [2]. The other cases were classified as "easy."

We trained models with a learning rate of 0.0001 using AdamW optimization, warm-up and linear scheduling. Our model was initialized with a pre-trained model of the cased Distilbert base [1] from the Hugging Face implementation [45].

We compared the performance of our model with state-of-the-art methods on ReferIt3D ([2, 40, 37, 38, 39]) based on the reported numbers on the challenge website [46] and corresponding papers. Since SAT [40] uses an extra 2D image dataset in their training, we separated it from non-SAT, their baseline model that is not trained with the extra dataset.

---

[1] Each value in $b_i$ is extended to 128-dimensions by PE and then concatenated to form a 768-dimensional vector.

| Dataset | Method | Overall | Easy | Hard | View-dep. | View-indep. |
|---|---|---|---|---|---|---|
| Nr3D | ReferIt3D [2] | 35.6 % | 43.6 % | 27.9 % | 32.5 % | 37.1 % |
| | ScanRefer [37] | 34.2 % | 41.0 % | 23.5 % | 29.9 % | 35.4 % |
| | InstanceRefer [38] | 38.8 % | 46.0 % | 31.8 % | 34.5 % | 41.9 % |
| | FFL-3DOG [39] | 41.7 % | 48.2 % | 35.0 % | 37.1 % | 44.7 % |
| | non-SAT [40] | 37.7 % | 44.5 % | 31.2 % | 34.1 % | 39.5 % |
| | Ours | **43.9** % | **51.0** % | **36.6** % | **41.7** % | **45.0** % |
| Nr3D w/ 2D images | SAT [40] | 49.2 % | 56.3 % | 42.4 % | 46.9 % | 50.4 % |
| Sr3D | ReferIt3D [2] | 40.8 % | 44.7 % | 31.5 % | 39.2 % | 40.8 % |
| | InstanceRefer [38] | 48.0 % | 51.1 % | 40.5 % | 45.4 % | 48.1 % |
| | non-SAT [40] | 47.4 % | N/A | N/A | N/A | N/A |
| | Ours | **56.0** % | **58.9** % | **49.3** % | **49.2** % | **56.3** % |
| Sr3D w/ 2D images | SAT [40] | 57.9 % | 61.2 % | 50.0 % | 49.2 % | 58.3 % |

Table 1: **Accuracy on ReferIt3D [2].** Our model outperformed state-of-the-art models on both Nr3D and Sr3D except for models with additional training data (SAT with 2D images). The average performance gap between ours and other models on Sr3D (10.6%) is larger than that on Nr3D (6.3%) since our model uses only spatial reasoning for the reference task.

## 4.3   Evaluation on ReferIt3D [2]

Table 1 shows the accuracy on Nr3D and Sr3D. Our model outperformed other models (without extra training datasets) on both Nr3D and Sr3D. It achieved 43.9% on Nr3D with an +8.3% improvement over the baseline method of ReferIt3D [2] and a 2.2% increase over the best accuracy from other models in Nr3D. For Sr3D, our model achieved 56.0%, which is a 15.2% and 8.0% increase over the baseline and the best accuracy in the Sr3D section, respectively. It is also comparable to the accuracy of SAT (with a 1.9% difference), which was trained with additional 2D image training data. These results prove that our model can accurately reason about spatial relationships from spatial-language embeddings and outperforms other models on both datasets despite the loss of information in appearance. In addition, less diverse language expressions and utterances only about spatial reasoning can explain our model's strong performance on Sr3D.

## 4.4   Evaluation with Ground-Truth Class Labels

We further investigated the model's spatial reasoning ability by removing noise in class labels. We replaced predicted class labels with ground-truth class labels, which was possible due to the explicit usage of class labels in our model.

Table 2 shows the result with and without ground-truth class labels. The first and second columns demonstrate the type of dataset in training and evaluation, respectively. For instance, the second row shows the evaluation result of the model trained with the Nr3D dataset with ground-truth class labels and evaluated on the Nr3D dataset with predicted class labels. When we trained and evaluated our model with ground-truth labels on Nr3D and Sr3D (row 4 and 8), we achieved 54.3% and 91.1% accuracy, respectively. Compared to the accuracy of models trained and evaluated with noisy labels, we realized an improvement is 10.4% and 35.1%, respectively, for each dataset. When we evaluated with ground-truth labels the models trained with noisy labels, their performance increase was 9.7% and 24.2%, respectively. Multiple reasons account for the difference in performance on Nr3D and Sr3D: diversity in the natural language dataset, a higher portion of view-dependent utterances or view-dependent utterances with no mention of orientation, and descriptions other than spatial relationships, such as color or appearance.

In addition, we examined the accuracy given switched datasets, namely, when we train a model on Nr3D and evaluate it using Sr3D, and vice versa. The last two rows in Table 2 show two switched evaluation results: 40.0% and 37.6% on Sr3D and Nr3D, respectively. The 37.6% accuracy on Nr3D from the model trained on Sr3D is comparable to the accuracy of non-SAT [40] and InstanceRefer [38]. It shows our model's generalizability of spatial reasoning and potential to transfer to different perception modules and datasets.

| Dataset | | Overall | Easy | Hard | View-dep. | View-indep. |
| --- | --- | --- | --- | --- | --- | --- |
| Training | Evaluation | | | | | |
| Nr3D-pred | Nr3D-pred | 43.9 % | 51.0 % | 36.6 % | 41.7 % | 45.0 % |
| Nr3D-gt | Nr3D-pred | 41.4 % | 48.0 % | 34.6 % | 38.1 % | 43.0 % |
| Nr3D-pred | Nr3D-gt | 53.6 % | 64.4 % | 42.5 % | 50.7 % | 55.0 % |
| Nr3D-gt | Nr3D-gt | 54.3 % | 65.5 % | 42.8 % | 49.1 % | 56.8 % |
| Sr3D-pred | Sr3D-pred | 56.0 % | 58.9 % | 49.3 % | 49.2 % | 56.3 % |
| Sr3D-gt | Sr3D-pred | 52.1 % | 53.8 % | 48.2 % | 43.9 % | 52.5 % |
| Sr3D-pred | Sr3D-gt | 80.2 % | 83.2 % | 73.1 % | 62.5 % | 81.0 % |
| Sr3D-gt | Sr3D-gt | 91.1 % | 93.1 % | 86.2 % | 67.0 % | 92.1 % |
| Nr3D-pred | Sr3D-pred | 40.0 % | 43.6 % | 31.5 % | 41.2 % | 39.9 % |
| Sr3D-pred | Nr3D-pred | 37.6 % | 45.3 % | 29.6 % | 34.5 % | 39.0 % |

Table 2: **Ablation with ground-truth class labels.** First and second columns show types of data used in training and evaluation. The high overall accuracy (91.1% at row 8) of the model both trained and evaluated with ground-truth class labels on Sr3D shows its spatial reasoning ability besides the perception noise. Accuracy gaps between ground-truth and predicted class labels on Nr3D and Sr3D (9.7%, 24.2%, respectively) indirectly tell us about language complexity and information loss due to classification. Transferring the model trained with Sr3D to Nr3D evaluation shows an overall number (37.6% at row 10) comparable to those from other methods. Our model can easily accommodate different classification models or datasets.

## 4.5 Ablation on Loss Terms

We trained models with different combinations of loss terms, and Table 3 shows the results. In addition to three tasks, we examined the effect of a text classification task that predicts the target class label $l \in \mathcal{L}$ from the tokens from utterance $X[M_U]$.

We found that only the binary classification loss shows its effect clearly (+3.5% from Ref. to Ref.-Clf., +3.9% from Ref.-Mask to Ref.-Mask-Clf., +1.0% from Ref.-Text to Ref.-Text-Clf.). The mask loss was not effective, and the text classification loss degraded accuracy. One hypothesis here is that the text classification loss does not help the model to transform the initial language embedding to a spatial-language embedding. The model may not require strong language constraints because (1) it already has a good initial language embedding, and (2) the text classification loss is independent of the scene's spatial configuration. Due to the performance drop, we chose our model without the text classification loss.

| Ref. | Clf. | Mask | Text | Overall | Easy | Hard | View-dep. | View-indep. |
| --- | --- | --- | --- | --- | --- | --- | --- | --- |
| ✓ | - | - | - | 40.6 % | 48.2 % | 32.8 % | 38.5 % | 41.6 % |
| ✓ | ✓ | - | - | 44.1 % | 51.3 % | 36.7 % | 41.0 % | 45.6 % |
| ✓ | - | ✓ | - | 40.0 % | 47.2 % | 32.5 % | 36.8 % | 41.5 % |
| ✓ | - | - | ✓ | 40.0 % | 48.7 % | 30.8 % | 37.8 % | 40.9 % |
| ✓ | ✓ | ✓ | - | 43.9 % | 51.0 % | 36.6 % | 41.7 % | 45.0 % |
| ✓ | ✓ | - | ✓ | 41.0 % | 49.7 % | 31.9 % | 39.3 % | 41.8 % |
| ✓ | ✓ | ✓ | ✓ | 40.0 % | 48.2 % | 31.5 % | 38.9 % | 40.6 % |

Table 3: **Ablation of loss terms on Nr3D.** The classification loss was effective, the mask loss did not significantly affect accuracy, and the text loss degraded accuracy. We chose the model without text losses (fifth row, in blue).

## 4.6 Viewpoint Annotation

View-dependent utterances without information about original viewpoint make the reference task in ReferIt3D [2] more challenging. For instance, utterances such as "*The door is wood with the handle on the left side.*" assume specific orientations of the agent, and it is impossible to recover the true orientation without knowing the referred object; this differs from view-dependent utterances with explicit viewpoint information, such as "*Facing the foot of the bed.*" However, the original dataset

| Correction | | Overall | Easy | Hard | View-dep. | View-indep. |
|---|---|---|---|---|---|---|
| Training | Evaluation | | | | | |
| ✓ | - | 43.5 % | 50.7 % | 36.0 % | 37.0 % | 46.6 % |
| ✓ | ✓ | 49.0 % | 56.0 % | 41.8 % | 54.4 % | 46.4 % |
| - | ✓ | 43.1 % | 50.3 % | 35.6 % | 40.4 % | 44.4 % |
| - | - | 43.9 % | 51.0 % | 36.6 % | 41.7 % | 45.0 % |

Table 4: **Comparison of accuracy with and without corrected orientations on Nr3D.**

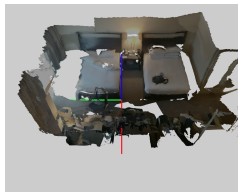 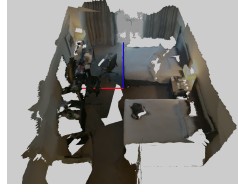 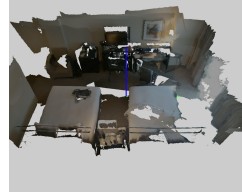 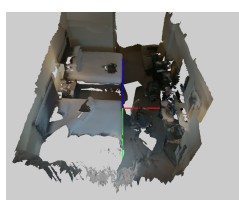

(a) An example of standard orientation 1.    (b) An example of standard orientation 2.    (c) An example of standard orientation 3.    (d) An example of standard orientation 4.

Figure 3: **Examples of standard orientations for viewpoint annotation on Nr3D (a-d).** We assume that the robot is always inside the room except for cases specified by utterances.

of ReferIt3D [2] does not distinguish the utterances without orientation information from those with it. Therefore, we split the view-dependent (VD) utterance category into two subcategories, VD-explicit and VD-implicit, where VD-explicit has explicit viewpoint information in the utterance. We then collected orientations from human annotators that validated the utterances. We set four standard orientations assuming the agent is in the room (around the center of the scene) and asked annotators to select all orientations that could be considered valid from the utterance. Figure 3 shows examples of the four orientations. We found that four orientations were sufficient to recover the original viewpoints of the speakers. In total, 12,680 view-dependent utterances of the Nr3D dataset were annotated; from these, 5,942 utterances were classified as VD-explicit. For train and test split, 10,206 and 2,474 utterances were annotated, respectively.

From the orientation for view-dependent utterances, we revised the dataset with the corrected orientation; we rotated the scene with respect to the annotation so all scenes remained valid at the canonical orientation. For view-independent utterances, we randomly rotated the scenes since they were valid in any direction. Table 4 shows the accuracy values for models trained with and without corrected orientations. At inference, we evaluated each model with and without corrected orientations, as well. The second row shows the accuracy of the model that was trained and evaluated with corrected orientations. Its overall accuracy was improved by $+5.1\%$ from the final model without the correction that was used in Table 1 (the last row in Table 4). Note that the improvement on view-independent utterances was marginal ($+1.4\%$), but the improvement on view-dependent ones was significant ($+12.7\%$). The first row shows the accuracy of a model trained with corrected orientations and evaluated on the test data without correction. This model achieved accuracy comparable to the final model ($-0.4\%$). This implies the correction helped the model to accurately interpret the view-dependent scene when the orientation was consistently aligned, introducing no unwanted bias.

## 5   Conclusion

We proposed LanguageRefer, a spatial-language model for 3D visual grounding in a reference task. LanguageRefer combines language embeddings from utterances and class labels with positional-encoded spatial information for efficient learning of the spatial-language embedding space without different modules for individual modality. Experimental results show that LanguageRefer outperformed state-of-the-art models on ReferIt3D with no additional training data. Analysis and ablations we performed demonstrate the effects of 1) noisy class labels, 2) arbitrary viewpoints in view-dependent utterances, 3) and the transfer of our model to different datasets for future robotics applications.

**Acknowledgments**

We would like to thank to Xiangyun Meng and Mohit Shridhar for discussion and feedback. This work is in part supported by NSF IIS 1652052, IIS 17303166, DARPA N66001-19-2-4031, DARPA W911NF-15-1-0543, Honda Research Institute as part of the Curious Minded Machine initiative, and gifts from Allen Institute for Artificial Intelligence.

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
