# OpenReview forum: "LanguageRefer: Spatial-Language Model for 3D Visual Grounding"
_robot-learning.org/CoRL/2021/Conference — CoRL2021 Poster_

### Official Review · Reviewer_PcsE · 2021-07-23

**Originality:** Very Good
**Technical Quality:** Good
**Clarity Of Presentation:** Good
**Impact:** 4

**Recommendation:**

Strong Accept: I recommend accepting the paper and will argue for my recommendation even if other reviewers hold a different opinion.

**Summary:**

This paper proposes a model to address the 3D visual grounding problem. Given a set of candidate 3D bounding boxes, the goal is to identify the object being referred to by a natural language expression. The proposed model combines spatial embeddings of the bounding boxes with a linguistic embedding from DistilBert. The resulting model is shown to outperform several baselines. In addition, the authors show that viewpoint-specific information is critical to grounding in 3D environments where the spatial information assumes a viewpoint for the speaker.


**Issues:**

The main issues with this work are missing details mentioned above. Adding these would greatly help the reader recognize the key contributions of this work.

**Reviewer Expertise:**

Very good: Comprehensive knowledge of the area

**Strengths And Weaknesses:**

One strength of this paper is the extensive set of comparisons and ablations, to which the proposed model achieves strong performance. The paper evaluates several design choices such as how important using ground truth class labels are and the relative importance of different components of the loss function. One particularly valuable insight is the importance of using a viewpoint that is consistent with viewpoint-dependent language. This is common for situated environments and is a valuable insight to the community.

The main weaknesses of this paper result from missing details needed to fully appreciate the take-aways of the proposed model. For example, I was not able to find descriptions of the baselines used in Table 1. This makes it difficult to state which component of the proposed model most contributed to the performance gain. It would be helpful to see these descriptions that highlight the key differences with the proposed model in Section 4.2 (or in the appendix if space does not allow). Relatedly, although the paper presents many interesting ablations, it would be additionally informative to compare the relative contributions of both the spatial encodings and the pretrained language model.

Additionally, the following points addressing the clarity of the paper would further strengthen the contribution:
- Visual information only factors into the prediction through class labels. This decoupling is shown to help the approach transfer to different datasets (Table 2). It would strengthen the comparison to show how a coupled perception-language model transfers between these datasets but perhaps this is outside the scope of this work. It is still an interesting result that the model performs well without further information and it would be useful to discuss any limitations that arise from this assumption.
- Regarding the loss function ablations, when the text loss and classification losses are left out, does the inference procedure described on line 171 change? Which combination of loss functions are used for the Table 1 evaluations?
- What is the value of “k” in the experiments? (Line 172)
- It would be helpful to describe the differences between the “Easy” and “Hard” splits in the result tables.


**Summary Of Recommendation:**

I recommend this paper be accepted to CoRL. The importance of viewpoint is an interesting contribution and I believe the identified issues can be addressed by the authors.

---

> ### Author Response · Authors · 2021-08-30
> **Response to Reviewer PcsE (1/2)**
>
> Thank you for your detailed and constructive feedback. We hope we addressed all the questions and issues you raised and we revised the paper in red according to your feedback. We have added the link to the webpage (https://sites.google.com/view/language-refer) at the end of abstract for providing more qualitative examples and visualization of the data.
>
> - **For example, I was not able to find descriptions of the baselines used in Table 1. This makes it difficult to state which component of the proposed model most contributed to the performance gain. It would be helpful to see these descriptions that highlight the key differences with the proposed model in Section 4.2 (or in the appendix if space does not allow).**
>
>     Thank you for your feedback. In section 2.2, a brief description of InstanceRefer, FFL-3DOG, and SAT were included. The description of SAT was in the footnote on page 3. We have added more highlights of our approach in comparison to previous approaches at the end of Section 2. Here is the copy of the explaining part from the paper: https://pastebin.com/AMYKQ4Bi.
>
> - **Relatedly, although the paper presents many interesting ablations, it would be additionally informative to compare the relative contributions of both the spatial encodings and the pretrained language model.**
>
>     Thank you for proposing this additional ablation study. We are currently training models 1) DistilBert [7] with a linear layer to form an embedding from 6D bounding-box information and 2) BERT [8] with positional encoding. We will update the result on the webpage (https://sites.google.com/view/language-refer) with this additional information as soon as we have the results.
>
> - **Visual information only factors into the prediction through class labels. This decoupling is shown to help the approaching transfer to different datasets (Table 2). It would strengthen the comparison to show how a coupled perception-language model transfers between these datasets but perhaps this is outside the scope of this work.**
>
>     Thank you for the interesting suggestion. We have looked at publicly released implementations of baselines, however, we could not find one to do such an experiment. However, we welcome any recommendations or pointers to do the experiment.
>
> - **It is still an interesting result that the model performs well without further information and it would be useful to discuss any limitations that arise from this assumption.**
>
>     Due to limited usage of perceptual information, our model is less accurate on utterances with color or shape descriptions. Our model takes an explicit class label in form of language as input and it may require extra annotations in order to transform other types of information into language. For instance, in order to provide a label 'red desk' from pointcloud of a desk instead of 'desk', we may need a special model or an extended model to describe color from the pointcloud. In addition, uncertainty in detection cannot be considered in our model as it only takes the finalized class labels.
>
>     To this end, extending the approach with rich attributes and studying object class uncertainty are considered as future work.
>
> - **Regarding the loss function ablations, when the text loss and classification losses are left out, does the inference procedure described on line 171 change? Which combination of loss functions are used for the Table 1 evaluations?**
>
>     The inference procedure remains the same regarding the loss function ablations. We only changed the training procedure to perform these ablations.
>
>     Thanks for pointing out the ambiguity. For the final model used for Table 1, we used reference, classification, and mask losses. To clarify this in the updated paper, we have highlighted the fifth row in Table 3.
>
> - **What is the value of “k” in the experiments? (Line 172)**
>
>     We used k=4 and this is added to the paper as well. Thank you for the feedback.
>
> - **It would be helpful to describe the differences between the “Easy” and “Hard” splits in the result tables.**
>
>     We have followed the definition of easy and hard cases from the ReferIt3D paper [2]; instances that contain more than two same-class distractors are defined as hard and the others are defined as easy. It is a scene-dependent category and not directly related to the language description. We added the explanation to the end of the first experiment setup paragraph in the paper.

---

> > ### Author Response · Authors · 2021-08-30
> > **Response to Reviewer PcsE (2/2)**
> >
> > [1] Vaswani, Ashish, et al. "Attention is all you need." Advances in neural information processing systems. 2017.
> >
> > [2] Achlioptas, Panos, et al. "Referit3d: Neural listeners for fine-grained 3d object identification in real-world scenes." European Conference on Computer Vision. Springer, Cham, 2020.
> >
> > [3] Chen, Dave Zhenyu, Angel X. Chang, and Matthias Nießner. "Scanrefer: 3d object localization in rgb-d scans using natural language." Computer Vision–ECCV 2020: 16th European Conference, Glasgow, UK, August 23–28, 2020, Proceedings, Part XX 16. Springer International Publishing, 2020.
> >
> > [4] Yuan, Zhihao, et al. "Instancerefer: Cooperative holistic understanding for visual grounding on point clouds through instance multi-level contextual referring." arXiv preprint arXiv:2103.01128 (2021).
> >
> > [5] Feng, Mingtao, et al. "Free-form Description Guided 3D Visual Graph Network for Object Grounding in Point Cloud." arXiv preprint arXiv:2103.16381 (2021).
> >
> > [6] Yang, Zhengyuan, et al. "SAT: 2D Semantics Assisted Training for 3D Visual Grounding." arXiv preprint arXiv:2105.11450 (2021).
> >
> > [7] Sanh, Victor, et al. "DistilBERT, a distilled version of BERT: smaller, faster, cheaper and lighter." arXiv preprint arXiv:1910.01108 (2019).
> >
> > [8] Devlin, Jacob, et al. "Bert: Pre-training of deep bidirectional transformers for language understanding." arXiv preprint arXiv:1810.04805 (2018).

---

> > > ### Comment · Reviewer_PcsE · 2021-08-30
> > > **Response to authors**
> > >
> > > I would like to thank the authors for taking the time to update the paper and respond to my comments. The additional details are helpful to understand the benefits of the proposed approach.
> > >
> > > My remaining confusion is regarding the loss ablations. Could the authors please clarify the relationship between the loss functions and the inference procedure? My current understanding was that the “text loss” was needed to train the transformer model to get the “predicted target class” mentioned in line 178 of the revised text. However, the revised text does not mention the “text loss” anymore. Where does the “predicted target class” come from in the overall pipeline that allows for this filtering during inference?

---

> > > > ### Author Response · Authors · 2021-08-31
> > > > **Response to Reviewer PcsE**
> > > >
> > > > Thanks for pointing out the ambiguity. At inference, we followed the approach proposed by the InstanceRefer [4] to reduce the number of objects in the reference task by filtering out objects that do not belong to the predicted target class (please refer to Figure 2 in InstanceRefer [4]). We employed an extra DistilBert-based target classification model (94 % accuracy) to get the predicted target class. Also, we extended the original approach by using top-$k$ predictions to prevent the filtering step from removing the true target instance. Any object whose top-$k$ class predictions (from the semantic classifier) do not contain the predicted target class is excluded from the reference task while including all the other objects. So the exclusion of text loss does not affect the prediction of the target class.
> > > >
> > > > We have updated the paper briefly pointing to this procedure that we adapted from InstanceRefer (line 178-182). Also, we have added Figure 6 on the website ([https://sites.google.com/view/language-refer#h.54l198l8mb7e](https://sites.google.com/view/language-refer#h.54l198l8mb7e)) about the inference procedure of the framework.

---

> > > > > ### Comment · Reviewer_PcsE · 2021-08-31
> > > > > **Response to authors**
> > > > >
> > > > > Thank you for the quick reply! The clarification and the figure are very helpful and address my confusion.

---

> > > > > > ### Author Response · Authors · 2021-08-31
> > > > > > **Response to Reviewer PcsE**
> > > > > >
> > > > > > We are glad that we addressed your confusion.
> > > > > >
> > > > > > Regarding your original suggestion on ablation, we have added a section on "Ablation of Positional Encoding and DistilBert" to the website: [https://sites.google.com/view/language-refer#h.njd4u3s5y5j](https://sites.google.com/view/language-refer#h.njd4u3s5y5j).

---

### Official Review · Reviewer_tKLL · 2021-07-23

**Originality:** Fair
**Technical Quality:** Excellent
**Clarity Of Presentation:** Very Good
**Impact:** 3

**Recommendation:**

Weak Accept: I recommend accepting the paper, but will not argue for my recommendation if the majority of other reviewers have a different opinion.

**Summary:**

This work develops a novel model for 3D visual grounding based on a 3D point cloud representation of a scene and a language utterance. This model simplifies earlier method by using pretrained language embeddings and a decoupled perception system, the method also also outperforms earlier methods on 2 datasets. Ablation studies motivate the design choices for this model also analyze the view dependence of the two Nr3D and Sr3D.

**Issues:**

* Some error analysis of baselines and LanguageRefer on the Nr3D dataset to show examples why LanguageRefer is qualitatively better, also some analysis of the remaining errors.
* Figure 2, there is a token "###acing" which is confusing, is this a typo?
* Numbers in tables would be more convincing if they had errors reported as well
* Table 4, mention the dataset in the caption for consistency.

**Reviewer Expertise:**

Good: General knowledge of the area

**Strengths And Weaknesses:**

Strengths:
* The transformer based, decoupled LanguageRefer model is well described and it is clear why all the components are needed.
* The ablations with perception noise help understand how much of the performance is purely from perception which may motivate further improvements
* Thorough description of experimental results with several baselines means the results are convincing.

Weakness
* The improvements on Nr3D seem convincing but are incremental, 41.7% vs 43.9%
* Some error analysis would help understand in which ways is LanguageRefer better than previous models? Are there any particular cases that it gets that InstanceRefer does not?
* The performance on Nr3D is still quite low, even with ground truth perception, why is this the case? What are some examples of errors that might motivate further improvements.

**Summary Of Recommendation:**

This work proposes an incremental improvement for 3D visual grounding with a simpler perception-decoupled transformer based architecture.

---

> ### Author Response · Authors · 2021-08-30
> **Response to Reviewer tKLL (1/2)**
>
> Thank you for your detailed and constructive feedback. We hope we addressed all the questions and issues you raised and we revised the paper in red according to your feedback. We have added the link to the webpage ([https://sites.google.com/view/language-refer](https://sites.google.com/view/language-refer)) at the end of abstract for providing more qualitative examples and visualization of the data.
>
> - **The improvements on Nr3D seem convincing but are incremental, 41.7% vs 43.9%**
>
>     We think this is mainly caused by the fact that our model ignores other meaningful information (e.g. color, shape) apart from semantic class label. This partially explains the differences in improvements between Nr3D and Sr3D.
>
> - **Some error analysis would help understand in which ways is LanguageRefer better than previous models? Are there any particular cases that it gets that InstanceRefer [1] does not?**
>
>     We tried to run InstanceRefer [1] code on ReferIt3D but they are not supporting ReferIt3D data but only ScanRefer [5], so we could not analyze cases and errors.
>
>     InstanceRefer has modules designed to extract embeddings from multiple contexts of the scene and a score for each object is computed by comparing its embedding to a language embedding. LanguageRefer is replacing designed modules with spatial-language embedding. Utterance also uses the same embedding space as a sequence of tokens. Then the reference task is done with Transformers considering all tokens from both language and objects together. We hypothesize that the shared and well-shaped spatial-language embedding space enables the Transformers to learn stronger attentions on both objects and tokens of utterances in contributing to the superior performance of LanguageRefer.
>
> - **Some error analysis of baselines and LanguageRefer on the Nr3D dataset to show examples why LanguageRefer is qualitatively better, also some analysis of the remaining errors.**
>
>     We add examples of prediction results from ReferIt3D and LanguageRefer on the website in the section Qualitative Comparison to ReferIt3D ([https://sites.google.com/view/language-refer#h.m8tlyc5vko4k](https://sites.google.com/view/language-refer#h.m8tlyc5vko4k)).
>
> - **The performance on Nr3D is still quite low, even with ground truth perception, why is this the case? What are some examples of errors that might motivate further improvements?**
>
>     First, we suspect that description about object appearances, such as color and shape, made the performance on Nr3D low. Our model is ignoring those attributes and even with the ground-truth perception, it does not have enough information for the task.
>
>     Second, the complexity of descriptions in Nr3D is high. Often they are lengthy, hard to interpret, and introducing a new concept, e.g. a group of objects. Some descriptions require guessing and verifying the original orientation since they are view-dependent but implicit about their original orientations.
>
>     A potential solution to this issue is to provide more data; our speculation is that the size of the data is too small relative to the diversity of the language expression. One qualitative example is illustrated in Figure 7C at the webpage (https://sites.google.com/view/language-refer#h.e307t2s0g3l3); among two tables in the scene, the model chose a wrong table from the description "table without any chairs around." The expression "without" seems to be rare in the dataset and it requires a different way of inference.
>
> - **Figure 2, there is a token "###acing" which is confusing, is this a typo?**
>
>     The token "###acing" comes from the tokenizer of DistilBert [2]. When representing a sentence as a sequence of tokens, there are multiple ways of building vocabulary. DistilBert chooses WordPiece [3] where it can produce sub-word tokens such as "###acing".
>
> - **Numbers in tables would be more convincing if they had errors reported as well**
>
>     We understand the concern of not including the errors. For faster and easier implementation, we stored the object detection result into a file and the identification result became static.
>
> - **Table 4, mention the dataset in the caption for consistency.**
>
>     Thank you for the feedback. We have updated the caption of Table 4 and Figure 3 so that readers can understand that it was done on Nr3D.

---

> > ### Author Response · Authors · 2021-08-30
> > **Response to Reviewer tKLL (2/2)**
> >
> >
> > [1] Yuan, Zhihao, et al. "Instancerefer: Cooperative holistic understanding for visual grounding on point clouds through instance multi-level contextual referring." arXiv preprint arXiv:2103.01128 (2021).
> >
> > [2] Sanh, Victor, et al. "DistilBERT, a distilled version of BERT: smaller, faster, cheaper and lighter." arXiv preprint arXiv:1910.01108 (2019).
> >
> > [3] Schuster, Mike, and Kaisuke Nakajima. "Japanese and korean voice search." 2012 IEEE International Conference on Acoustics, Speech and Signal Processing (ICASSP). IEEE, 2012.
> >
> > [4] Feng, Mingtao, et al. "Free-form Description Guided 3D Visual Graph Network for Object Grounding in Point Cloud." arXiv preprint arXiv:2103.16381 (2021).
> >
> > [5] Chen, Dave Zhenyu, Angel X. Chang, and Matthias Nießner. "Scanrefer: 3d object localization in rgb-d scans using natural language." Computer Vision–ECCV 2020: 16th European Conference, Glasgow, UK, August 23–28, 2020, Proceedings, Part XX 16. Springer International Publishing, 2020.

---

### Official Review · Reviewer_rNyP · 2021-07-23

**Originality:** Good
**Technical Quality:** Very Good
**Clarity Of Presentation:** Very Good
**Impact:** 4

**Recommendation:**

Weak Accept: I recommend accepting the paper, but will not argue for my recommendation if the majority of other reviewers have a different opinion.

**Summary:**

This paper presents a model for grounding referring expressions to a 3D point cloud scene. Authors take the approach of encoding the language component and the the 3D scene and decode the referring object using a combination of losses. The 3D scene encoder uses a standard model for predicting the 3D bounding boxes and a classification label. The proposed approach treats the classification outputs as a sentence and determines a language-encoding for the same. The positional encoding is then encoded as a spatial embedding which is later combined with the language embedding prior to decoding the object labels. The model uses a combination of loss functions to arrive at the target object. Authors present thorough experiments on a number of benchmark data sets.

**Issues:**

- The approach directly processes the predicted classification labels and bounding boxes without explicitly modeling uncertainty in the classification labels or the geometry of the bounding boxes.
- How extensible is the approach to richer inter-object relations. For example, "bring me the cup in the microwave" where an object can contain another object referred to by the instruction.

**Reviewer Expertise:**

Good: General knowledge of the area

**Strengths And Weaknesses:**

Strengths
- The authors address an important problem in human-robot interaction.
- The model uses a realistic 3D point cloud data as input.
- The approach also shows robustness to the orientation of the person providing the instruction.




**Summary Of Recommendation:**

The paper contributes an approach for grounding language understanding in 3D scenes. The results are comprehensive.

---

> ### Author Response · Authors · 2021-08-30
> **Response to Reviewer rNyP**
>
> Thank you for your constructive feedback and questions. In addition to the discussion, we have added the link to the webpage ([https://sites.google.com/view/language-refer](https://sites.google.com/view/language-refer)) at the end of abstract for providing more qualitative examples and visualization of the data.
>
> - **The approach directly processes the predicted classification labels and bounding boxes without explicitly modeling uncertainty in the classification labels or the geometry of the bounding boxes.**
>
>     Thank you for raising an interesting issue. This can be the limitation of our model in that it relies on finalized perception results and cannot handle the uncertainty in perception. One of our motivations is to examine the spatial inference capacity of the model and to give a clear distinction from the perception error where it used to be combined with other factors in previous works. By using the perfect perception module, we can see the limit of the model and, by comparing the performance with and without the perfect perception module, we can indirectly measure the effect of perception noise. As shown in Table 2, we observe a large accuracy gap (+35.1 % in Sr3D, +10.4 % in Nr3D) when we assume a perfect perception module.
>
> - **How extensible is the approach to richer inter-object relations? For example, "bring me the cup in the microwave" where an object can contain another object referred to by the instruction.**
>
>     Thank you for another interesting question. We believe the model has the potential to infer the state of containment once it has embeddings from positions and sizes. Our proposed model has the capacity of discriminating the referred object from other same-class instances by interpreting expressions such as 'in front of', 'left', 'right'. We can consider containment can be another case of spatial relationship.
>
>     When it comes to applying this to an embodied system, we may need to consider temporal object permanence and strategies to extract correct bounding-boxes. We think it is a very interesting direction of research and we hope our approach would be applicable to potential research with richer inter-object relationships.
>
> [1] Yang, Zhengyuan, et al. "SAT: 2D Semantics Assisted Training for 3D Visual Grounding." arXiv preprint arXiv:2105.11450 (2021).

---

### Official Review · Reviewer_3bBQ · 2021-07-24

**Originality:** Good
**Technical Quality:** Very Good
**Clarity Of Presentation:** Good
**Impact:** 3

**Recommendation:**

Weak Accept: I recommend accepting the paper, but will not argue for my recommendation if the majority of other reviewers have a different opinion.

**Summary:**

For a given 3D point cloud of a scene, 3D bounding boxes, and language instructions, the method aims to identify the target object from a set of potential candidates. To achieve this, four modules (classification, spatial embedding, language embedding, and spatial-language model) have been used in the system. The method was evaluated on the ReferIt3D dataset (containing Sr3D and Nr3D datasets), and the results show that the suggested system outperforms the baseline models. Further, the experiments present that the method performs better for the expressions with spatial references (Sr3D) than natural references(Nr3D), which include the color features or appearance of the objects.

**Issues:**

- The paper does not emphasize enough what has been achieved in the 3D visual grounding and what are the contributions of the suggested system over existing approaches. The previous approaches on comprehending referring expressions in 3D aimed to identify the target object among candidate objects, as in this paper. It would be nice to focus more on the novel aspects of the suggested system.
- What is the number of object classes that can be identified by the semantic classifier? Can it classify all the objects categories in the ReferIt3D dataset?
- It is not clear how the spatial embedding is obtained in the system. The paper states that positional encoding PE(·) is employed to transform the bounding box vector, but it is not straightforward to understand how PE(·) converts bounding box vectors to spatial embedding.
- The target classification loss is defined as L_tar, but it is used as L_clf for the rest of the paper.
- Did you fine-tune the baseline methods (i.e., the ones used for comparison in Table 1) on the ReferIt3D dataset?
- What are the easy and hard categories reported in the tables? It has been forgotten to explain in the paper.
- The authors state that the model trained with corrected orientations achieved comparable accuracy to the best model even without corrections at inference (+0.4%, −0.4%). Which columns of Table 4 were compared to obtain these values?
- Figure 2 is not easy to follow. Spatial embedding representation might be positioned closer to DistilBert to reduce the length of arrows.

**Reviewer Expertise:**

Very good: Comprehensive knowledge of the area

**Strengths And Weaknesses:**

Strengths:
- The paper addresses a critical topic and different important subtopics. It focuses on understanding language instructions in 3D. It further discusses the importance of viewpoint orientations and the impacts of perception noise in spatial reasoning tasks. These issues have been less focused in computer vision studies, and they are critical in robotics applications.
- The system is evaluated in detail.  It is compared with five previous approaches for Nr3D and four existing approaches for Sr3D. Further ablations, additional results with ground-truth classes, and more evaluations with view-dependent and independent expressions are also presented in the experiments section.

Weaknesses:
- The suggested system is complex, composed of many different modules, and used four different losses. In this case, failure in one module can mislead the main prediction and might cause identifying the wrong object as a target. Further, in ablation studies, the paper shows that the mask loss does not affect the accuracy much, and the text loss decreased the accuracy. Therefore, it is not clear why these components are suggested as parts of the proposed system.
- The method focuses on understanding spatial references, and it mainly considers the positional embeddings. Accordingly, it obtains lower accuracies in the Nr3D dataset than the Sr3D dataset since the Nr3D dataset contains more features such as color and shape. However, in many robotics applications, only spatial relations might not be enough to disambiguate the target objects, and users might use varying features while describing the target objects. Such a system might fail when an object is described only with its appearance.

**Summary Of Recommendation:**

Although the suggested system is complex and mostly focuses on spatial references, it includes critical analyses about viewpoint orientations and perception noise, and these are crucial aspects for robots to build the bridge between the natural language and vision in real-world environments (see strengths and weaknesses for details).

---

> ### Author Response · Authors · 2021-08-30
> **Response to Reviewer 3bBQ (1/2)**
>
> Thank you for your detailed and constructive feedback. We hope we addressed all the questions you raised and we revised the paper in red according to your feedback. In addition, we have added the link to the webpage (https://sites.google.com/view/language-refer) at the end of the abstract for providing more qualitative examples and visualization of the data.
>
> - **The suggested system is complex, composed of many different modules, and used four different losses. In this case, failure in one module can mislead the main prediction and might cause identifying the wrong object as a target.**
>
>     We agree that the failure in the former module can lead to the error of the final identification. While decoupling perception (object detection) is a limitation, our modular approach is transferable and enables us to study the approach more thoroughly (ablation studies). Additionally, just like our model uses the pretrained language model from extremely large language data outside of the task of interest, the modular approach may leverage extra data to improve its performance in perception.
>
> - **Further, in ablation studies, the paper shows that the mask loss does not affect the accuracy much, and the text loss decreased the accuracy. Therefore, it is not clear why these components are suggested as parts of the proposed system**.
>
>     Thanks for pointing this out. We did not use the text loss in our final model. Our formulation (Equation 1) and the system diagram (Figure 2) described in the submission had all the loss terms used in our experiments. We understand this must be confusing. Hence we updated the paper by dropping the text loss term in Equation 1 and updated the corresponding parts in the paper.
>
>     For the mask loss, we keep the term because all the ablation experiments were done with the setup. We believe the effect of the term is limited.
>
>     We observed that the losses on the language description did not contribute much to the performance. However, the embeddings from the language model are crucial for the Transformers to properly attend to the utterances and objects in the scene to perform the reference task.
>
> - **The paper does not emphasize enough what has been achieved in the 3D visual grounding and what are the contributions of the suggested system over existing approaches. The previous approaches on comprehending referring expressions in 3D aimed to identify the target object among candidate objects, as in this paper. It would be nice to focus more on the novel aspects of the suggested system.**
>
>     Thank you for your suggestion. We added contribution and comparison of our approach over existing approaches at the end of Section 2. Hope this helps readers to understand the novel aspect of our approach.
> Here is the copy of the explaining part from the updated paper: https://pastebin.com/AMYKQ4Bi.
>
> - **What is the number of object classes that can be identified by the semantic classifier? Can it classify all the objects categories in the ReferIt3D dataset?**
>
>     In the original dataset, the total number of object classes was 607 and we reduced the total number of object classes to 110. There are 76 target object classes out of 607 and we keep those classes the same. During the process of reducing the number of classes, we observed some patterns in the classes.
>
>     Among classes, we observed class pairs of the same category with singular and plural forms, e.g. 'kitchen cabinet' vs 'kitchen cabinet***s***'. Plural forms of classes were often used to indicate a group of objects; we have decided to keep their original forms.
>
>     In addition, a large part of non-target classes have small numbers of instances in the data and they are often too fine-grained. For instance, we had a lengthy list of classes containing the word 'chair':
>     `['chair', 'armchair', 'folded chair', 'stack of folded chairs', 'recliner chair', 'stack of chairs', 'rocking chair', 'sofa chair', 'massage chair', 'beanbag chair', 'office chair', 'folded chairs'].`
>     A few variations of 'folded chair' are also considered as distinct classes. However, vocabulary in language descriptions was less diverse than the diversity of fine-grained classes. Therefore, we reduced the number of classes. After the reduction, the classes with 'chair' are reduced to:
>     `['chair', 'armchair', 'sofa chair', 'office chair'].`
>     Reducing the number of classes increased the average classification accuracy and help to improve the referring task accuracy. It can be thought of as a trade-off between the expressiveness of the classes versus the accuracy of classification.

---

> > ### Author Response · Authors · 2021-08-30
> > **Response to Reviewer 3bBQ (2/2)**
> >
> > - **It is not clear how the spatial embedding is obtained in the system. The paper states that positional encoding PE(·) is employed to transform the bounding box vector, but it is not straightforward to understand how PE(·) converts bounding box vectors to spatial embedding.**
> >
> >     We take 6D values of a bounding box, (x, y, z, w, h, d), and apply sinusoidal spatial functions to increase the dimension from each scalar value from 1 to 128. Then spatially-encoded vectors are concatenated to form a 768-dimensional vector (6 x 128) and added to the BERT embedding. (The computation is done in `pe_from_tensor` from `utils.positional_encoding` in the code we provided.) We use the sinusoidal functions used in [1]. We added more explanation of the positional encoding to the updated paper and omitted citation (we only had the citation in Figure 2.)
> >
> > - **The target classification loss is defined as L_tar, but it is used as L_clf for the rest of the paper.**
> >
> >     Thank you for the correction. We replaced $L_{tar}$ with $L_{clf}$.
> >
> > - **Did you fine-tune the baseline methods on the ReferIt3D dataset?**
> >
> >     The numbers of the baseline methods are reported in the ReferIt3D challenge and we took the numbers for granted ([https://referit3d.github.io/benchmarks.html](https://referit3d.github.io/benchmarks.html)). At the time of submission, TransRefer3D did not release their paper, so we had to omit their numbers. We added the explanation of the source of numbers to the paper.
> >
> > - **What are the easy and hard categories reported in the tables?**
> >
> >     We have followed the definition of easy and hard cases from the ReferIt3D paper [2]; instances that contain more than two same-class distractors are defined as hard and the others are defined as easy. We added the explanation to the end of the first experiment setup paragraph in the paper.
> >
> > - **The model trained with the corrected orientations achieved comparable accuracies to the best model, but where is it in Table 4?**
> >
> >     The first row in Table 4 describes the accuracy of a model trained with training data whose orientations are corrected and evaluated on the test data whose orientations are not corrected. This model achieved comparable accuracy to the final model (trained on data without correct orientations) when tested on data without corrections to the orientation (4th rows in Table 4).
> >
> >     In other words, what we wanted to describe here is that the correction to the orientation can be considered as removing noises from the data. Let us define the original data D and the data with corrections to the orientation D'. Since we are removing noises from the data, a model trained with D', clean data, will not work well when evaluated with the noisy data D. Otherwise, the model trained with the noisy data, D, and evaluated on D and D' does not show a large gap since the model is already trained with noisy data.
> >
> > - **Figure 2 is not easy to follow. Spatial embedding representation might be positioned closer to DistilBert to reduce the length of arrows.**
> >
> >     Thank you for your advice. We moved the spatial embedding representation towards the DistilBert [7] and changed the orientations of the arrows from circles as representations of objects for clear distinction from arrows to a caption.
> >
> > [1] Vaswani, Ashish, et al. "Attention is all you need." Advances in neural information processing systems. 2017.
> >
> > [2] Achlioptas, Panos, et al. "Referit3d: Neural listeners for fine-grained 3d object identification in real-world scenes." European Conference on Computer Vision. Springer, Cham, 2020.
> >
> > [3] Chen, Dave Zhenyu, Angel X. Chang, and Matthias Nießner. "Scanrefer: 3d object localization in rgb-d scans using natural language." Computer Vision–ECCV 2020: 16th European Conference, Glasgow, UK, August 23–28, 2020, Proceedings, Part XX 16. Springer International Publishing, 2020.
> >
> > [4] Yuan, Zhihao, et al. "Instancerefer: Cooperative holistic understanding for visual grounding on point clouds through instance multi-level contextual referring." arXiv preprint arXiv:2103.01128 (2021).
> >
> > [5] Feng, Mingtao, et al. "Free-form Description Guided 3D Visual Graph Network for Object Grounding in Point Cloud." arXiv preprint arXiv:2103.16381 (2021).
> >
> > [6] Yang, Zhengyuan, et al. "SAT: 2D Semantics Assisted Training for 3D Visual Grounding." arXiv preprint arXiv:2105.11450 (2021).
> >
> > [7] Sanh, Victor, et al. "DistilBERT, a distilled version of BERT: smaller, faster, cheaper and lighter." arXiv preprint arXiv:1910.01108 (2019).
> >
> > [8] Devlin, Jacob, et al. "Bert: Pre-training of deep bidirectional transformers for language understanding." arXiv preprint arXiv:1810.04805 (2018).

---

> > > ### Comment · Reviewer_3bBQ · 2021-09-02
> > > **Post-Rebuttal**
> > >
> > > I would like to thank the authors for addressing my comments in detail. The paper is improved in the revised version. Dropping the text loss helps to simplify the model, and the additional information is helpful to emphasize the novelty.

---

### Meta-Review · Area_Chair_V77A · 2021-08-14

**Recommendation:** Accept (Poster)
**Confidence:** 4

**Metareview:**

All reviewers agree that the visual grounding is an important and interesting topics and that the proposed approach, taking advantage of 3D visual data and language instructions, is promising.

However, the paper could make an even better case for the proposed method by improving clarity and by discussing the robustness, scalability and the limitations of the approach in further detail.

## Post-Rebuttal Update
The authors addressed and clarified the concerns raised by the reviewers, improving the paper in the process.

---

> ### Author Response · Authors · 2021-08-30
> **Response to Area Chair V77A**
>
> We appreciate the area chair and the reviewers for their feedback and time. We found the feedback helpful and responded to the questions from each reviewer in comments and revised the paper highlighted in red.

---

### Decision · Program_Chairs · 2021-09-13

**Decision:**

Accept (Poster)

**Comment:**

All reviewers agree that the visual grounding is an important and interesting topics and that the proposed approach, taking advantage of 3D visual data and language instructions, is promising.

However, the paper could make an even better case for the proposed method by improving clarity and by discussing the robustness, scalability and the limitations of the approach in further detail.

## Post-Rebuttal Update
The authors addressed and clarified the concerns raised by the reviewers, improving the paper in the process.